# Rate-Distortion Performance and Incremental Transmission Scheme of Compressive Sensed Measurements in Wireless Sensor Networks

**DOI:** 10.3390/s19020266

**Published:** 2019-01-11

**Authors:** Felipe da Rocha Henriques, Lisandro Lovisolo, Eduardo Antônio Barros da Silva

**Affiliations:** 1Coordenação de Telecomunicações—CEFET/RJ—Campus Petrópolis, Petrópolis 25620-003, Brazil; 2PROSAICO–DETEL/PEL–FEN, Universidade do Estado do Rio de Janeiro, Rio de Janeiro 20559-900, Brazil; lisandro@uerj.br; 3PEE/COPPE, Universidade Federal do Rio de Janeiro, Rio de Janeiro 21941-901, Brazil; eduardo@smt.ufrj.br

**Keywords:** wireless sensor network, compressive sensing, successive approximation

## Abstract

We consider a Wireless Sensor Network (WSN) monitoring environmental data. Compressive Sensing (CS) is explored to reduce the number of coefficients to transmit and consequently save the energy of sensor nodes. Each sensor node collects *N* samples of environmental data, these are CS coded to transmit M<N values to a sink node. The *M* CS coefficients are uniformly quantized and entropy coded. We investigate the rate-distortion performance of this approach even under CS coefficient losses. The results show the robustness of the CS coding framework against packet loss. We devise a simple strategy to successively approximate/quantize CS coefficients, allowing for an efficient incremental transmission of CS coded data. Tests show that the proposed successive approximation scheme provides rate allocation adaptivity and flexibility with a minimum rate-distortion performance penalty.

## 1. Introduction

A Wireless Sensor Network (WSN) is an ad hoc network whose nodes have four basic units: Sensing, processing, communication, and power (often comprising just a battery). Each node operates as long as it has energy [1]. In order to increase sensor nodes autonomy, methods leading to energy saving in WSN have appeared [2]. Transmission bandwidth is tightly associated with energy consumption—the larger the bandwidth the higher the energy consumption [1].

A signal is said to be sparse if there is a base in which it can be represented with only a few non-zero coefficients. Compressive Sensing (CS) [3] explores sparsity to reduce the number of coefficients required to represent a given signal within some reconstruction fidelity. In CS, few coefficients are obtained through linear measurements (projections), and the original signal can be inferred from them by means of a linear program or greedy algorithm [4]. This works if the quantity of measurements is a bit larger than the signal sparsity (which can be much lower than its *Nyquist* limit [3]).

In a WSN, a sink node receives traffic from several sensor nodes, and usually has much more computational power than these other nodes, that must be simple and be low power consuming. Such a scenario fits the CS paradigm: Simple encoding and complex decoding [5]. In this work, the sparsity of collected signals is explored by means of CS to reduce the quantity of transmissions in WSNs, saving sensor nodes energy.

Notwithstanding CS remarkable signal encoding capability, CS decoding has a phase change behavior: If the quantity of measurements is sufficiently large then a reasonable signal estimative (in terms of mean squared error) is possible; otherwise signal is incorrectly reconstructed. This hampers rate-distortion management by controlling the quantity of coefficients and one must rely mostly on coefficients quantization for rate-distortion control. Therefore, we investigate an incremental quantization scheme for CS coefficients to provide a finner rate-distortion control. In addition, this scheme allows the sensor nodes to first encode signals with a coarse quality and let the sink (the application layer) to ask for extra bits to improve the signal, without requiring significant extra sensor node resources.

In summary, we investigate CS-coded transmissions in order to save sensor nodes energy. Real-life signals are used in simulations and distinct recovery methods are also investigated. The robustness of CS-coded framework against packets loss is verified, and a novel successive approximation scheme for energy conservation of sensor nodes is proposed.

After this introduction, Section 2 presents a brief literature review on techniques for energy saving in WSN. This allows to better situate our work within the field and discuss our objectives and contributions. Section 3 brings the Compressive Sensing fundamentals and the reconstruction methods investigated and, within it, Section 3.2 presents the problem model and employed notation. Section 4 discusses the experimental methodology and presents results on the rate-distortion performance and energy consumption of quantized CS measurements encoding in WSN. Section 4.4 presents the impact of packet losses in the rate-distortion performance. The incremental transmissions scheme, using successive approximation, is presented in Section 5. Then, conclusions are presented in Section 6.

## 2. Energy Conservation Schemes in WSN

Several energy conservation/saving schemes have been proposed for WSNs. WSN-based signal acquisition using CS is discussed in [6]. The correlations among signals detected by different sensors are explored to further reduce the amount of samples required for reconstruction. Environmental data gathered by a WSN located in the Intel Berkeley Research Lab [7] are considered in [6]. Spatial correlation of the sensed signal is explored in [8] in order to propose a compressive-based sleeping WSN, in which a TDMA scheme is employed imposing only a fraction of the sensor nodes to be active in each time slot. Data gathered by the active nodes are transmitted to a base station and data corresponding to the sleeping nodes are estimated from others.

A cluster-based energy-efficient data collection by using the CS framework is proposed in [9]. The WSN is divided in clusters using two distinct communication models: In the first one, sensor nodes send their measurements to the cluster head that forwards the CS measurements directly to a Base Station; the second method uses intermediate cluster heads to forward CS measurements to the base station. Authors calculate the power consumption for both methods.

A belief propagation algorithm [10] is considered in [11] to reconstruct CS measurements sent by sensor nodes, and iterative message passing is used to find the solution. A star topology is considered, in which sensor nodes directly send their CS measurements to the sink.

WSN employ digital communication, consequently coefficients derived from sensor measurements must be quantized before transmission. However, the analyses performed in [6,8,9,11] disregard this aspect, assuming that coefficients are densely quantized. Some insight on the quantization of compressive sensed measurements is presented in [12,13,14]. In [12], the 1-bit Compressive Sensing framework is proposed, which preserves only the sign information of each random measurement and the signal is recovered within a scale factor. This 1-bit CS framework is applied to data gathering in WSNs in [15].

In [13], an empirical analysis of the rate-distortion performance of CS is presented, in the context of image compression. The average distortion introduced by quantization on CS measurements is studied in [14] by comparing two reconstruction algorithms, the Basis Pursuit [16] and the Subspace Pursuit [17] CS recovery schemes. In [18], a CS-based coding technique that considers a binned progressive quantization framework in order to overcome the poor performance of CS applied to image compression is presented. The decoder firstly generates an initial reconstructed image, then quantization refinement occurs in order to subsequently improve the reconstructed image. The finer the quantization of the coefficients, the more bits are required for their transmission. In the present work, the amount of energy spent for transmission is considered a critical issue since one desires to consume the less energy while guaranteeing the best representation of the monitored signal; and both aspects will depend upon the CS-coefficients quantization scheme.

Authors in [19] address the dimensionality reduction problem, applied to hyperspectral images, by using manifold learning methods. They propose the UL-Isomap scheme, in order to solve the main problems of the LIsomap method [20]: The shortcoming of random landscape and the slow computation speed of LIsomap. A Vector Quantization is considered to improve the landmark selection, and the computational speed of LIsomap scheme is accelerated by using random projections, in order to reduce the bands of HSI data. In [21], authors consider the nonlinear dimensionality reduction problem, also applied to hyperspectral image classification, and propose an Enhanced-Local Tangent Space Alignment (ENH-LTSA) method. As the work in [19], this proposed scheme applies random projections to reduce the dimension of HSI data, improving the construction of the local tangent space. It aims to reduce the computational complexity of the LTSA method [22].

Compressive Sensing scheme also considers a random projection framework, since the signal to be compressed is projected in the sensing matrix to generate *M* CS coefficients. However, unlike [19,21], we apply the CS scheme to compress environmental data gathered by a WSN. Furthermore, our work aims to investigate some network aspects, such as the energy consumption of sensor nodes and the impact of packets loss in the reconstruction of the monitored signals.

A simple way to mitigate transmission losses (or packet erasures) is by using *acknowledgments* (ACK) [23]. When a transmission is received, the sink may send an ACK, confirming the receipt of the measurement. If the sensor node receives the ACK, then the subsequent data can be sent. Otherwise, a retransmission is required. We consider a WSN without ACKs, since to retransmit packets means greater energy consumption. Authors in [24] investigate the characteristics of lossy links in WSNs, and apply the CS framework to overcome this problem. Experimental results show that the WSN can transmit with high quality, while reducing energy consumption because of the CS characteristics, i.e., if there is a sufficient number of received packets and if the sink is capable of identifying which coefficients (packets) were lost. The transmission over a noisy channel and the evaluation of the reconstruction error in the presence of packet erasures while saving sensor nodes energy, are considered in [25]. Two approaches for CS are studied in [25]: A cluster-based method that considers aggregation to compress data, and a consensus-based scheme in order to estimate the linear projections of sensor nodes; a basis pursuit denoising (BPDN) algorithm [16] is used for reconstruction by the sink node.

In order to save sensor nodes energy in a surveillance application using a Visual Sensor Network (VSN), a routing framework called PRoFIT is proposed in [26]. In this VSN application scenario, sensing nodes divide images into bit-planes [27]. There are two layers that are created for captured images. The first one contains the most significant bit-planes (transmitted with higher priority) and the second layer bares less significant bit-planes (transmitted with lower priority). From the first layer, the image is reconstructed at the sink with a certain degree of detail, so that some immediate action can be taken based on the image content. If a more detailed reconstruction is required, the sink uses the second layer with remaining bit-planes. This strategy can be understood as a successive approximation scheme [28], in which bit-planes are incrementally transmitted, in order to provide a better reconstruction.

### Objectives and Contributions

In this paper we investigate a quantized CS framework, aiming at reducing the amount of transmissions in a WSN. In doing so, one expects that network nodes save energy, thus increasing WSN autonomy. We consider different environmental data: temperature, humidity, and illumination [7]. For reconstruction, we investigate the performance of three distinct CS reconstruction schemes. We also evaluate the impact of packet loss in the rate-distortion performance of that encoding scheme.

As the transmitted CS measurements are quantized, the rate-distortion behavior of the reconstruction of monitored signals must be evaluated. This is done considering an exhaustive combination of uniform quantizers with distinct amounts of CS coefficients. The analysis generates an RD operational curve that goes over the best compromises between rate and distortion, allowing to reduce energy consumption in transmissions (bits, packets, and bursts) carried out by each sensor node, while keeping the rate-distortion performance sufficiently close to the optimal.

We propose to employ quantization refinements to incrementally transmit CS measurements, resulting in a successive approximation framework for CS. Consequently, at any operational RD point, an increment in rate is mapped into how many extra coefficients the sensor shall transmit or how many extra bits shall be used to improve previously transmitted CS measurements or both simultaneously. In practice, this scheme allows sensor nodes to first encode signals with a coarse quality and let the sink (the application layer) to ask for refinements without requiring significant extra sensor node resources, thus providing a sink-based control of the RD operational point of sensor nodes. It is worth to mention that while the RD control is coordinated by the sink, each sensor can be at different operational points. To the best of our knowledge, this is the first time that the incremental CS quantized measurements is discussed.

## 3. Compressive Sensing Framework

Let a signal $x\in R^{N}$ (a column vector) be sparse in some domain and s be its sparse representation (that is with S<<N non-null coefficients)
(1)s=Ψx,
in which $\Psi \in R^{N\times N}$ is a transform that provides a sparse s, for example, DCT or Wavelet. Consider that
(2)y=Φx,
in which $y\in R^{M}$ (M<N) is the vector that contains the coefficients (linear measurements) and $\Phi \in R^{M\times N}$ is called *measurement (or sensing) matrix* [29]. The *m*-th coefficient in y is obtained by the inner product between x and the measurement function $\phi_{m}$ (the *m*-th row of *sensing matrix*). Given y, a reconstruction procedure should aim at finding the solution for $y=\Phi \Psi^{*}s$, with the smallest $l_{0}$ norm for s (the number of its non-zero entries). ($\Psi^{*}$ is the complex conjugate transpose of Ψ.) Unfortunately, this is an ill-posed problem requiring a combinatorial approach [3].

Alternatively, in order to circumvent the combinatorial problem, one minimizes the $l_{1}$ norm of the reconstructed signal [4]. In this case, the optimization problem is given by
(3)$$\hat{s}=\arg\min_{\begin{array}{c} s \end{array}}{∥s∥}_{1}\;s.t.\;y=\Phi \hat{x},\;\hat{x}=\Psi^{*}\hat{s},$$
where ${∥s∥}_{1}=\sum_{i=1}^{n}\left|s_{i}\right|$ is the $l_{1}$ norm of the signal s, being $s_{i}$ the components of s.

### 3.1. Reconstruction Methods

There are some convex optimization algorithms that can be used to solve (Equation 3). Following, three distinct recovery methods considered in this work are briefly mentioned: The Newton algorithm with a log-barrier method, used in the L1-magic MatLab toolbox [30]; the greedy-based algorithm called A*OMP [31]; and the shrinkage-based algorithm called Least Absolute Shrinkage and Selection Operator (LASSO) [32].

#### 3.1.1. Newton + Log-barrier

The recovery procedure aims at minimizing the $l_{1}$ norm of s with quadratic constraints, i.e.,
(4)$$\hat{s}=\arg\min_{\begin{array}{c} s \end{array}}{∥s∥}_{1}\;s.t.\;{∥y-\Phi \Psi^{*}\hat{s}∥}_{2}\leq E,$$
in which E is an empirically chosen parameter. This optimization is done by recasting the problem in Equation (Equation 4) as a *Second-Order Cone Program* (SOCP) [30]. Then, a Newton algorithm with a logarithmic barrier (log-barrier) method [33] is used to solve the optimization problem in Equation (Equation 4). The recovery procedure is implemented in the L1-magic package [30]. In [34], authors consider the algorithm in L1-magic to reconstruct CS-coded environmental signal (temperature) in a WSN. The advantage of using the *Newton + log-barrier* method is that it directly minimizes the L1-norm of the sparse signal, which is small with respect to their energy.

#### 3.1.2. A* Orthogonal Matching Pursuit

The A*OMP is an algorithm based on the *Orthogonal Matching Pursuit* (OMP) algorithm [35], that uses atoms from a dictionary to expand x. At each iteration, the expansion uses the dictionary atom having the largest inner-product with the residue. After each iteration, the orthogonal projection of the residue onto the selected atoms is computed. At iteration *l*, the set of atoms $A_{l}$ selected from Φ for representing y incorporates the dictionary atom that best matches $r^{l-1}$, doing
(5)$$\begin{array}{cc} s & =\arg\min_{\begin{array}{c} V_{n}\in \Phi \end{array}}\langle r^{l-1},V_{n}\rangle , \end{array}$$
(6)$$\begin{array}{cc} A_{l} & =A_{l}\cup s, \end{array}$$
(7)$$\begin{array}{cc} c & =\arg\min_{\begin{array}{c} \tilde{c}\in R^{l} \end{array}}{∥y-A_{l}\tilde{c}∥}_{2}, \end{array}$$
(8)$$\begin{array}{cc} r^{l} & =y-A_{l}c, \end{array}$$
in which $V_{n}\in R^{N}$ and n=1,2,⋯,N are dictionary atoms (columns of the dictionary Φ), c is the vector of the coefficients and $r^{l}$ is the residue after the *l*-th interaction. At the end, $A_{l}$ contains the support of x (the original signal to be coded), and c contains their nonzero entries (that define the sparsity of the signal).

The A*OMP considers the best-first search tree [36], in which multiple paths can be evaluated during the search to improve reconstruction accuracy. The search is called A* and looks for the sparsest solution [31]. The *A*Search* gives an advantage to the A*OMP against the OMP algorithm, smaller average run-times [31]. Authors in [6] considered MP-based algorithms to reconstruct environmental signals, such as temperature, humidity, and illumination.

#### 3.1.3. LASSO

The *Least Absolute Shrinkage and Selection Operator* (LASSO) [32] aims at minimizing the residual sum of squares, subject to the sum of the absolute values of the coefficients being less than a constant. The reconstruction of CS measurements by LASSO can be expressed as
(9)$$\arg\min_{\begin{array}{c} x \end{array}}{\;∥\Phi x-y∥}_{2}\;s.t.\;{∥x∥}_{1}\leq T,$$
in which T is a nonnegative scalar.

LASSO was used to reconstruct temperature and humidity CS-coded signals in [37]. Moreover, since LASSO minimizes the sum of square errors, it is a well-suited method to recover noisy data [38]. In this work, we implemented LASSO using the SPGL1 software packet [39].

### 3.2. Using CS in WSNs

We employ CS to encode and decode environmental data monitored by a WSN. Figure 1 depicts the basics of the coding/decoding process:**Sensing**—The WSN node $S_{i}$ makes measurements (from the environment), and stores them in a vector $x_{i}\in R^{N}$
(10)$$x_{i}=\left[x_{i}\left[1\right],x_{i}\left[2\right],\cdots ,x_{i}\left[N\right]\right].$$**Compression**—The compressed sensing vector $y_{i}\in R^{M}$ (M<N),
(11)$$y_{i}=\left[y_{i}\left[1\right],y_{i}\left[2\right],\cdots ,y_{i}\left[M\right]\right],$$
is obtained from $x_{i}$ using (Equation 2).**Quantization**—For transmission, the compressive sensed coefficients are quantized using a uniform scalar quantizer of bit-depth *B* (that is, with $2^{B}$ reconstruction levels), this produces the quantized coefficients vector $y_{q_{i}}$.**Transmission**—Each node forwards its quantized coefficients vector $y_{q_{i}}$ to the sink node.**Reception**—Due to possible packet loss, the received coefficients vector at the sink node is
(12)$$y_{r_{i}}=\left[y_{r_{i}}\left[1\right],y_{r_{i}}\left[2\right],\cdots ,y_{r_{i}}\left[L\right]\right]$$
with L≤M.**Reconstruction**—The $y_{r_{i}}$ is used at the sink to reconstruct a version $\hat{x_{i}}$ of the original signal.

## 4. Rate-Distortion Analysis of Using CS Encoding in WSN

Each quantizer bit-depth provokes a different quantization error, with different impacts in the reconstructed signal. In quantized CS, two non-linear operations occur: Measurements quantization and signal reconstruction; differently from the transform-quantization-inverse transform approach. Therefore, for a fair rate-distortion performance analysis one may look for an empirical exam [13]. For that, a range of CS measurement quantities (*M*) and different quantizers (having as design parameter the quantizer bit-depth *B*, resulting in $2^{B}$ quantization levels) are employed. The experiments employ the signals from the Intel Berkeley Research lab [7], in which 54 sensor nodes were spread inside the lab sensing environmental data for more than one month.

### 4.1. Simulation Set-Up and Rate-Distortion Criteria

First, we consider a one-hop WSN, and use real temperature, humidity, and illumination signals, gathered by the 54-node WSN located in the Intel Berkeley Research Lab [7]. For the CS framework, it is assumed that the sensed signal is sparse in the DCT domain [6] and for sensing matrix construction, *M* random waveforms of an N×N Noiselet [40] are used.

The *M* quantized CS coefficients are assumed to be entropy encoded [41]. Therefore, it is reasonable to assume that the average rate spent to encode each signal sample is
(13)$$R=\frac{M}{N}\times H\left(Y_{q}\right),\;\mathrm{bits}/\mathrm{coeff},$$
where $H(Y_{q})$ denotes the entropy of the quantized coefficients set $Y_{q}$, which in our experiments is empirically obtained, and *M* is the quantity of transmitted coefficients per signal block of length *N*. This model (Equation (Equation 13)) is adopted in this work because it can be used to compare the rate for different *M*, *N* and quantizer bit-depths *B*.

Distortion is defined in terms of the Normalized Mean Squared Error (NMSE)
(14)$$\mathrm{Normalized}\;\mathrm{MSE}\left(dB\right)=10\log_{10}\left(\frac{E\left[{\left(x_{i}-{\hat{x}}_{i}\right)}^{2}\right]}{\left|\right|x_{i}{\left|\right|}^{2}}\right).$$

In Equation (Equation 14), E[·] is the expected value operator.

### 4.2. Rate—Distortion Results

Figure 2 presents the RD performance of the reconstruction of the temperature signal using L1-magic (in the recovery procedure for the L1-magic algorithm, the E parameter used to solve (Equation 4) was empirically chosen. We highlight that the RD results to be presented are averaged over 300 runs (using different seeds). For each number of measurements and bit-depth (*M*,*B*), E was varied and the E value that generates the lowest distortion was employed.), A*OMP and LASSO. The dimension of the signal block is set to N=512. We consider quantizers bit-depths (B) of {4,6,8,10} and M∈{8,16,32,64,128,256} CS measurements. For each reconstruction algorithm and quantizer bit-depth *B* we present the RD curve. One observes that as rate increases (in bits/sample, in this case due to augmenting *M*), in general there is an improvement in the reconstruction of the signal (decrease of the normalized MSE) for the three recovery algorithms. This expected behavior is due to the more CS measurements used in the reconstruction. These results also show that smaller reconstruction errors are obtained as the bit-depths of the quantizers increase.

L1-magic presented a better RD performance (that is, smaller distortion for the same rate) than both A*OMP and LASSO did. However, L1-magic did not obtain a monotonically decreasing distortion in function of rate. Such behavior has been observed in various simulation settings, and it was also observed to be sensitive on E (Equation (Equation 4)). The A*OMP and LASSO did not show such non-monotonic behavior. It also can be verified that the LASSO presents a better RD performance than the A*OMP.

Consequently, we employ the LASSO algorithm to make a more detailed evaluation of the RD performances of the reconstructions of temperature, humidity, and illumination signals. We generate denser operational RD curves than the ones in Figure 2. The number of CS measurements ranges from M=8 to M=256, in steps of 2; and the bit-depths (*B*) varied within the set {4,5,6,7,8,9,10}. As in the previous results, N=512. Figure 3 shows the RD curves of the reconstruction of the temperature (a), humidity (b), and illumination (c) signals, for each bit-depth (there, we present only the rate-distortion curves for 4, 6, 8, and 10 bits (similar behaviors are observed for quantization using 5, 7, and 9 bits)).

The graphs in Figure 3 also present the convex hull [42], which is composed by the points that lead to the best trade-off between rate and distortion, that is, corresponding to the best quantizers for each rate. The RD convex hull (all quantizers were considered in the computation of the convex hull, i.e., B∈ {4,5,6,7,8,9,10}) represents the empirically optimal RD operational curve.

Considering the same sampling rate, the complexity of CS tends to increase with *N* (above we considered N=512). Measurements are inner products [43] and the quantity of required arithmetic operations per measurement is proportional to the block length *N*. However, increasing *N* may help to better exploit signal sparsity. This motivates investigating the impact of the length of the signal block (*N*) in the rate-distortion performance. The number of CS measurements ranges from M=8 to M=256, in steps of 2 and the bit-depths (*B*) are within the set {4,5,6,7,8,9,10}. We also consider three different signal block lengths, N={128,256,512}. The reconstruction was obtained with LASSO. Figure 4 presents the results obtained for this experiment. One can observe that the RD performance improves as *N* increases. However, for the envisioned scenario the trade-off between the length of the signal block and the energy consumption of sensor nodes should be considered instead.

### 4.3. Energy Consumption Evaluation of a Multi-hop CS-based WSN: A Case Study

In this experiment, we consider that a multi-hop fifteen-nodes WSN is monitoring a temperature signal. A sensor node, located in the position of node $S_{47}$ from [7], (39.5,14) in meters, collects temperature samples of the environment and transmits the measurements to the sink node ($S_{0}$), located in the (0.5,17) coordinates (these coordinates are in meters and relative to the upper right corner of the lab.) using multiple hops. Since we consider a multi-hop WSN, the other thirteen nodes are used as routers to forward the packets from the sensor to the sink. In each simulation run, the position of the thirteen routers are drawn from the remaining nodes. We have performed 10 simulation runs.

TrueTime 1.5 [44] is used to perform the simulations, the IEEE 802.15.4 standard [45], referred to as ZigBee and widely used in WSNs, is considered for communication between nodes and the AODV (*Ad hoc On-Demand Distance Vector*) routing protocol is used by routing nodes. We consider a state-based energy model for the sensor node. Roughly, each node can perform some basic tasks when active: To measure, to process, to transmit, or to receive data. Each one of these operation modes is associated to a specific energy consumption, based on the time period in which the sensor node stays in those modes. For details please refer to [46], in which authors proposed DECA, a Distributed Energy Conservation Algorithm for WSN. Sensor nodes that run DECA at their application layer consider the known past (previously measured and transmitted samples) to predict the instant of next transmissions, by using a linear predictor. Then, sensor nodes save energy by managing the necessity for transmissions, based on the variation rate of the monitored signal. The sink uses a first-order interpolator to reconstruct the original signal from the received data.

Considering the transmission of a temperature signal (N=512) using M=14 entropy-coded-CS measurements quantized using *B* = 4, 6, 8, and 10 bits, Table 1 shows the transmission rate (in bits/sample), the NMSE (in dB) and the energy (in Joules) consumed by the sensor node (transmitter). As expected, one can observe that the distortion (NMSE) decreases as the rate increases (in bits per sample). On the other hand, more energy is consumed by sensor nodes, since more bits are transmitted by entropy-coded-CS measurements. This clearly shows the interdependence among quantization, distortion, and energy consumption.

We now compare the CS framework against DECA [46] with respect to the energy consumption and the NMSE of the reconstructed signal. Figure 5 shows the NMSE (in dB) of the reconstruction of N=512 samples of a temperature signal and the energy consumption (in Joules) of the sensor node (transmitter) after the transmission of the *M* measurements/coefficients. Black curves show the results for CS framework, varying the amount of transmitted measurements (*M*), and each point presents the NMSE value for B=4, B=6, B=8, and B=10 bits. The red curve shows the results for DECA, with the points for δ=1%, δ=0.1%, and δ=0.01%. The δ parameter is a threshold imposed by DECA for the maximum relative error in the reconstruction of the original signal. Smaller values of δ tend to improve the reconstruction of the monitored signal (with the decrease in the NMSE), since more transmissions are required. Furthermore, DECA uses 8 bits for each transmitted measurement.

It can be observed that CS provides a better reconstruction of the monitored signal for most of the results of Figure 5. This is so because there is a specific recovery algorithm (LASSO) used in the CS scheme, while in DECA the sink uses a simple first-order interpolator to reconstruct the received signal. Regarding to the energy consumption, the smaller the value of δ, the higher the number of transmissions, leading to a high energy expenditure when DECA is used. On the other hand, when sensor node uses the CS framework, the *N* acquired samples have to be stored before the transmissions, consuming energy.

### 4.4. Impact of Coefficient (Packet) Loss

So far, we have considered that a sensor node $S_{i}$ transmits its quantized measurements $y_{q_{i}}=\left[y_{q_{i}}\left[1\right],y_{q_{i}}\left[2\right],\cdots ,y_{q_{i}}\left[M\right]\right]$ and that all are available for signal reconstruction. However, as previously discussed it is inherent to WNS that a part of the transmitted coefficients be lost. Consequently, the sink node receives *L* measurements (L≤M<N), i.e., $y_{r_{i}}=\left[y_{r_{i}}\left[1\right],y_{r_{i}}\left[2\right],\cdots ,y_{r_{i}}\left[L\right]\right].$

If each entry $y_{q_{i}}\left[k\right]$ is transmitted within a data packet, in the case one packet is lost then only one measurement is missed. Once transmitting a measurement within a packet, each packet loss results an erased coefficient [47]. In [48], orthogonal projections are used in the transmitter for compensating erasures, however [48] considers a transmitter-aware scheme, in which the occurrence of erasures is known by the transmitter in order to recover erasure errors by subspace projection—which may be hard to implement. Several transmission protocols include sequence numbers in the packets; this is indeed the case for the IEEE 802.15.4 standard (which we consider) [49,50]. The sequence numbers allow to reorder segments in the transport layer and to identify lost frames in the link layer. As a result, in the CS-encoded WSN monitoring framework, at the sink one can easily prune the sensing matrix rows corresponding to lost coefficients (lost packets) before reconstruction.

For the experiments, we consider that the WSN is monitoring temperature, and that N=512 blocks are transmitted using varying bit-depths (*B*) and number of coefficients (*M*). The number of CS measurements used ranges from M=8 to M=256, in steps of 2; and bit-depths *B* vary within the set {4,5,6,7,8,9,10}. Different packet loss ratios are considered. Figure 6 shows the resulting RD performances for some of them, along with the optimal RD curve (obtained from the convex hull of the rate-distortion points presented in Section 4.2, which corresponds to 0% packet loss ratio). Figure 6 presents the curves for 10%, 30%, and 50% packet loss ratios (similar results were obtained for 20% and 40% packet loss ratios.), which are averages on 300 random runs.

To evaluate the impact of the packet loss, we employ the Bjontegaard Delta ΔBD—a metric widely used in audio and video coding in order to evaluate average RD performance differences between two encoders [51,52]. Table 2 shows the ΔBD between the ideal case (no-losses or 0% packet loss ratio) and the cases undergoing different packet loss ratios; the ΔBD of the Normalized Mean Squared Error (NMSE) between the curves in Figure 6 increases as more packet are lost, however, in any case the performance deterioration is very mild as reported by the ΔDB of the NMSE being smaller than 2 dB.

The ability of the CS framework to perform reasonably as far as enough coefficients are used to encode the signal translates into resilience against packet loss (coefficient erasures). This produces the behavior in Figure 6 and Table 2. From an energy-efficiency point of view, this is significant, since retransmissions can be avoided without largely impacting signal quality. The presented results corroborate the “democracy property” of CS, which states that all measurements contribute with the same amount of information [53]. If all CS measurements contribute with the same amount of information and if the sink knows which coefficients are missing, then one simply ignores them when reconstructing the signal.

## 5. Successive Approximation Using CS-WSNs

So far, we have used CS to code the signal captured by WSN nodes allowing to configure the RD operating point—(M,B). However, it could be necessary to improve the reconstructed signal at the sink node, even at expenses of more bits. One could retransmit the signal at a higher rate, and this would waste resources. On the other hand, some form of scalability could be pursued. We propose to do that using an incremental transmission framework for CS-Coded WSN-signals. This guarantees good RD performance while economizing resources, and, consequently, saving sensor energy. Extra-bits are transmitted to refine already known quantized CS coefficients or convey unknown coefficients leading to an improve in the signal reconstruction in an incremental fashion. This strategy keeps RD performance as good as possible, since any extra bit transmitted by nodes is used in the reconstruction and not to replace previously known ones. It is straightforward to see that this strategy allows to save energy in sensor nodes.

We employ a successive approximation scheme such that nodes may incrementally transmit CS measurements. Let a given node to code a set of $M_{i}$ measurements using $B_{i}$ bits and transmit them, spending a total of $R_{i}=M_{i}B_{i}$ bits; this defines the operational point $\left(M_{i},B_{i}\right)$ in the M×B space. Suppose now that the sink node (decoder) needs more information, in order to generate a better version of the reconstructed signal, subject to an increment of $\Delta R_{i}$ bits. The increment must improve RD performance while remaining as close as possible to the optimal RD points. In any case, $\Delta R_{i}$ corresponds to the selection of a new operational point $\left(M_{i}+\Delta M_{i},B_{i}+\Delta B_{i}\right)$. The $\Delta B_{i}$ and $\Delta M_{i}$ correspondent increase in the quantity of transmitted bits is
(15)$$\Delta R_{i}=M_{i}\Delta B_{i}+\Delta M_{i}B_{i}+\Delta M_{i}\Delta B_{i},$$
where $M_{i}$ and $B_{i}$ are known. $\Delta R_{i}$ may be seen as the target increment in rate, demanded by the sink, and Equation (Equation 15) presents the mapping from the target to the increments in the quantities of bits and measurements leading to a new operational point.

It is desirable for the sequence of operational points resulting from extra transmissions to provide a path along the optimal RD curve. Therefore, each $\left(M_{j},B_{j}\right)=\left(M_{i}+\Delta M_{i},B_{i}+\Delta B_{i}\right)$ should satisfy
(16)$$\left(M_{j},B_{j}\right)=\min_{\begin{array}{c} {k\in K|}_{M_{i},B_{i}} \end{array}}D\left(M_{k},B_{k}\right),$$
where D(·) is the distortion function while K is the set of new candidates—Equation (Equation 16) means that, from $\left(M_{i},B_{i}\right)$, one must search for $\left(M_{j},B_{j}\right)$ leading to the smallest distortion among all K candidates. Figure 7 illustrates that idea. It presents an optimal coding path (sequence of encoder setups) on the M×B plane, the $\left(M_{j},B_{j}\right)$ pairs having the best RD trade-of. It depicts the convex hull of the RD of the temperature signal, with the number of measurements (*M*) ranging from 8 to 256, in increments of 2, and quantized bit-depths values (*B*) varying within the set {4,5,6,7,8,9,10,11,12,13}, for a signal block of length N=512, this considers the NMSE as distortion criterion.

### 5.1. Moving along (M,B) Pairs

Increments in *M* and *B* may occur in several ways: the number of measurements increases while bit-depth is unaltered, the coefficients quantization may be refined while number of measurements is unaltered, or both are incremented simultaneously. The first two possibilities correspond to paths along horizontal or vertical directions on the M×B plane, respectively. In the more general case, instead, both *M* and *B* could change jointly, moving the coder setup up and to the right in Figure 7. Therefore, starting from a given $\left(M_{i},B_{i}\right)$ operational RD point, the coder should move to a new $\left(M_{j},B_{j}\right)$ pair leading to the minimal distortion for the resulting increment in the rate.

Each operational point represents a specific CS-quantized coding setup on the M×B plane. The quantizers are designed such that their input-output mappings intersect. The quantization rule using B+1 bits refines the quantization rule for *B* bits. Thus, when moving from the operational point *i* to the operational point i+1 just the refinements of previously transmitted values need to be transmitted, i.e., $B_{i+1}-B_{i}$ bits per value.

The initial RD-optimal setup point $\left(M_{0},B_{0}\right)$ is defined from the initial rate. The $M_{0}$ measurements are quantized and transmitted with $B_{0}$ bits. If there is demand for more bits, then a new RD-optimal point is to be used. The new point $\left(M_{1},B_{1}\right)$ may be selected from an operational setup table according to the desired rate increase. That is, for an operational point $\left(M_{i},B_{i}\right)$ there may be several possible operational setups to use, the one to be used $\left(M_{i+1},B_{i+1}\right)$ will depend on the actual increase in the quantity of transmitted bits subject to distortion reduction. Once that is decided, the sensor sends the $M_{i+1}-M_{i}$ extra measurements quantized with $B_{i+1}$ bits and the extra $B_{i+1}-B_{i}$ bits refining the already transmitted $M_{i}$ measurements.

Nevertheless, one must still compute the operational point to which the sensor node must shift. A sensor node can not do that on-line, as its energy would be exhausted due to the extensive required computations. Alternatively, a training set can be employed to analyze the RD performance and generate a table indicating the best shift at a given operational point and the desired increment in the quantity of transmitted bits. That data can be used in two different fashions: sensor nodes can be aware of the table and fetch the new operational point for a given rate increment; or, alternatively, the sink may inform the sensor node of the new operational point; the later reduces the requirements on sensor node memory size and marginally increases the data sent by the sink to request the relevant bits for signal reconstruction improvement.

### 5.2. Successive Approximation Table Construction

Given that $M_{i}$ coefficients have been already transmitted, one can define the set of new possible quantities of CS coefficients as ${\left\{M_{k}\right\}}_{k\in K}$, $M_{k}>M_{i}$. From Equation (Equation 15), one can compute the corresponding possible bit-depths ${\left\{B_{k}\right\}}_{k\in K}$, for a predefined target increment $\Delta R_{i}$. Dividing both sides of Equation (Equation 15) by $\Delta M_{i}=M_{k}-M_{i}$, one obtains
(17)$$\begin{array}{cc} \frac{\Delta R_{i}}{\Delta M_{i}}=\frac{M_{i}\Delta B_{i}}{\Delta M_{i}}+B_{i}+\Delta B_{i},\mathrm{thus} & \Delta B_{i}=\frac{\frac{\Delta R_{i}}{\Delta M_{i}}-B_{i}}{1+\frac{M_{i}}{\Delta M_{i}}}. \end{array}$$

Consequently, to build the successive approximation encoding table (the one that provides the increments $\Delta M_{i}$ and $\Delta B_{i}$ for moving to the next RD-optimal point subject to the operational point ($M_{i},B_{i}$) increment in rate $\Delta R_{i}$), one may use Algorithm 1. In this procedure, $\left(M_{i},B_{i}\right)$ is the current RD-optimal pair of node $S_{i}$; $d_{R}$ is the proportional increment in the rate; $c_{n}$ is the amount of $\left(M_{i},B_{i}\right)$ points generated in each operational curve; $d_{N}$ is a factor that limits the amount of operational curves that are tested. The value of $c_{n}$ defines the possible increments in $M_{i}$, stored in $\Delta M_{j}$. For each of them, the correspondent bit-depth increase is computed in lines 7–9. The best choice among them (leading to the same increment $\Delta R_{j}$ bits in rate) is evaluated in the eleventh line. This provides an “incremental path” (or successive approximation) for encoding and transmission of CS quantized measurements. If this process is performed off-line, it does not impact the energy consumption of the sensor nodes; however, sensor nodes either need to be aware of the table or be informed about the new operational setup.

Table construction complexity is controlled by $d_{R}$ and $c_{n}$, that limits the quantity of CS reconstructions that are evaluated for the training data. The input factor $d_{R}$ defines the rate increment that is evaluated while $c_{n}$ defines the number of possible operational points to be tested at a given ($M_{i},B_{i}$) to choose the next operational point $\left(M_{i+1},B_{i+1}\right)=\left(M_{i}+\Delta M_{i},B_{i}+\Delta B_{i}\right)$. In other words, $c_{n}$ defines the amount of candidate setups for the given percent increment in rate $d_{R}$. Note that the larger $c_{n}$ is, the more trials are required, i.e., more ($\Delta M_{i}$, $\Delta B_{i}$) are generated and analyzed. Suppose that one starts at a rate $R_{i}$ and wants to increment it to $R_{i}+\Delta R_{i}$, as $d_{R}$ increases, the sensor traverses less operational points to go from a rate $R_{i}$ to a rate $R_{i}+\Delta R_{i}$. Therefore, $d_{R}$ restricts the amount of generated curves that have to be tested. The smaller that $d_{R}$ is, the more precisely the rate can be adjusted, at expenses of a more complex training process.

**Algorithm 1:** Construction of the successive approximation table   **input**: *N*, dR, dN, cn   **output**: (Mj,Bj)
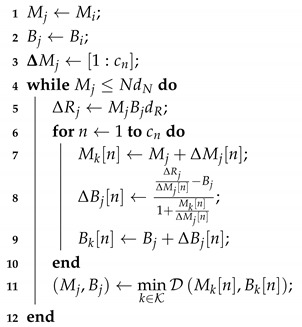


To illustrate the procedure, assume that the initial RD-optimal point is $\left(M_{0},B_{0}\right)=\left(8,4\right)$, the next point (the first refinement point) is chosen from at most ten (M,B) pairs, i.e., $c_{n}=10$, for a fixed rate increment defined by $d_{R}$. Each one of the 10 candidates produces a distortion. After testing, the one leading to the minimal distortion is chosen—Equation (Equation 16). This is the next operational point, for example, $\left(M_{1},B_{1}\right)=\left(10,5\right)$. At this point, the same process applies, resulting in the operational point $\left(M_{2},B_{2}\right)=\left(13,6\right)$, for example. The procedure is iterated until $M\leq Nd_{N}$, restricting the maximum number of measurements as a fraction of the signal block length *N*, $\left(d_{N}<1\right)$.

### 5.3. Simulation Results

#### 5.3.1. Rate-Distortion Evaluation

We consider temperature, humidity, and illumination signals, gathered by a WSN in the Intel Berkeley Research Lab [7]. In this setup, we consider a block length of N=512 samples and LASSO for reconstruction. From each $\left(M_{i},B_{i}\right)$ point, we consider rates increments of 10%, i.e., $d_{R}=0.1$. We also impose $d_{N}=0.5$ ($M\leq \frac{N}{2}$) and $c_{n}=10$. The Bjontegaard Delta metric (ΔBD) is used to evaluate the RD performance of the successive approximation scheme against the exhaustively generated optimal RD curve. The dataset employed to obtain the table mapping the successive approximation setups is different from the one employed to evaluate the RD performance.

Figure 8 presents the RD-optimal convex hull curve (as in Section 4.2) and the successive-approximation scheme rate-distortion curve for the reconstruction of the temperature (a), humidity (b), and illumination (c) signals. The Bjontegaard Delta between the exhaustively generated RD-optimal curves and the successive approximation RD performances for temperature, humidity, and illumination signals are ΔBD=0.32 dB, ΔBD=0.72 dB, and ΔBD=0.67 dB. The results indicate that the successive-approximation scheme operational-RD curve closely matches the RD-optimal convex hull for the three monitored signals. Nevertheless, for each one, the incremental-transmission scheme operational curves contain more RD points than the convex hull does, meaning that it allows a finer adjustment of the rate.

#### 5.3.2. Energy Consumption Evaluation: Case Studies

We now empirically evaluate the possible energy savings derived from the use of the proposed successive approximation scheme at sensor nodes. In the experiment, we evaluate the energy consumed when a sensor node traverses its operational curve, moving on a sequence of operational points $\left(M_{j},B_{j}\right)$. We compare the energy expenditure when the proposed successive approximation mechanism is employed against simply transmitting the whole data using one encoder with the operational set-up leading to the same total.

In this experiment (Scenario 1) we consider the same set-up used in Section 4.3. Table 3 presents the results. The first column shows the operational points traveled by the sensor (the first encoder configuration (RD operational point) is $\left(M_{0},B_{0}\right)=\left(8,4\right)$. At each operational point, the following one is obtained considering $c_{n}=10$ and $d_{R}=0.1$.). The second column presents the energy required to transmit the $M_{j}$ CS measurements using $B_{j}$ bits without using the proposed scheme. The third column presents the energy consumed when the successive approximation scheme is employed. The last column in Table 3 shows the percentage reduction in energy consumption gained by using the proposed scheme. The reduction comes from having to transmit only extra measurements and refinement bits. This energy economy behavior is steadily observed for all operational points.

Now we consider another operational curve to be traversed by sensor nodes. The same fifteen-node WSN used in simulations of the previous scenario are also considered for Scenario 2, but in this case all sensor nodes traverse their operational curve transmitting to the sink $M_{j}$ CS coefficients, each one with $B_{j}$ bits, using the proposed successive approximation scheme. Results are shown in Table 4, presenting the average MSEs and the average energy consumptions for all network nodes. One observes the same energy economy behavior as in Scenario 1, for all operational points.

A disadvantage of the proposed successive approximation scheme is the fact that sensor nodes are unable to perform an on-line construction of their operational curves (successive approximation tables). This is so because of the extensive required computations to generate the operational points. Thus, sensor nodes must be aware of the table or the sink must inform the new operational point. However, results presented in Figure 8, and Table 3 and Table 4 show that the proposed scheme provides a rate-distortion performance that closely matches the convex hull, while sensor nodes save their energy.

## 6. Conclusions

In this work, the quantization of compressive sensed measurements in WSNs was considered. The resulting rate-distortion performance using three distinct reconstruction methods was analyzed for temperature, humidity, and illumination data. RD analyses were presented for a different quantity of CS measurements and quantizer bit-depths (*M* and *B*, respectively) using for signal reconstruction the $l_{1}$ norm minimization, A*OMP, and LASSO algorithms. The best results were obtained with the first one, and LASSO outperformed the A*OMP recovery algorithm. In the tests, other features of CS-based WSN data encoding were evaluated as the capability to save sensor nodes energy and robustness against packet losses.

We proposed an scheme to incrementally transmit CS encoded sensor nodes measurements. The proposed scheme refines the data available at the sink upon its demand in terms of fractional rate increments. The scheme designs superimposed quantization rules for the measurements, allowing to reuse previously transmitted values as the quantization intervals get slender. Using this scheme, sensor nodes need only to transmit quantization refinements of already partially known measurements. This allows to save sensor nodes energy. Experiments show that the proposed scheme has a rate-distortion performance that closely matches the optimal RD curve.

It shall be highlighted that the presented strategy goes beyond choosing a CS-encode-quantize operational points, since we devised a strategy to shift between operational points in order to obtain a scalable bit-stream. In doing so, the proposed scheme allows for reconstructing the sensor data under an RD trade-off that includes the inherent non-linearities of both CS recovering algorithms and quantization.

## Figures and Tables

**Figure 1 sensors-19-00266-f001:**
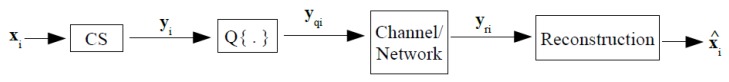
Compressive sensing (CS) coding/decoding in the wireless sensor network (WSN) scenario.

**Figure 2 sensors-19-00266-f002:**
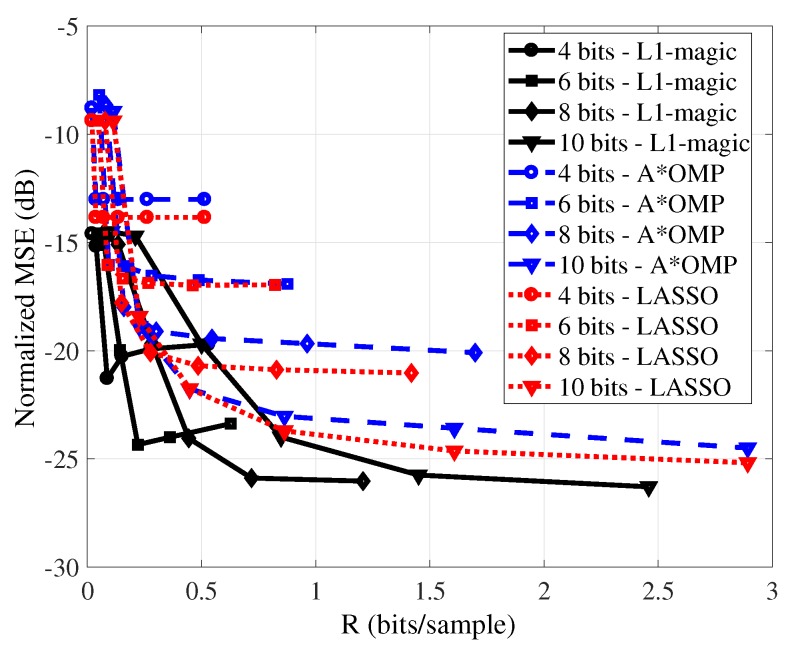
Rate-distortion (RD) results for a temperature signal for several quantization bit-depths using L1-magic, A*orthogonal matching pursuit (OMP), and least absolute shrinkage and selection operator (LASSO). The block length *N* is equal to 512.

**Figure 3 sensors-19-00266-f003:**
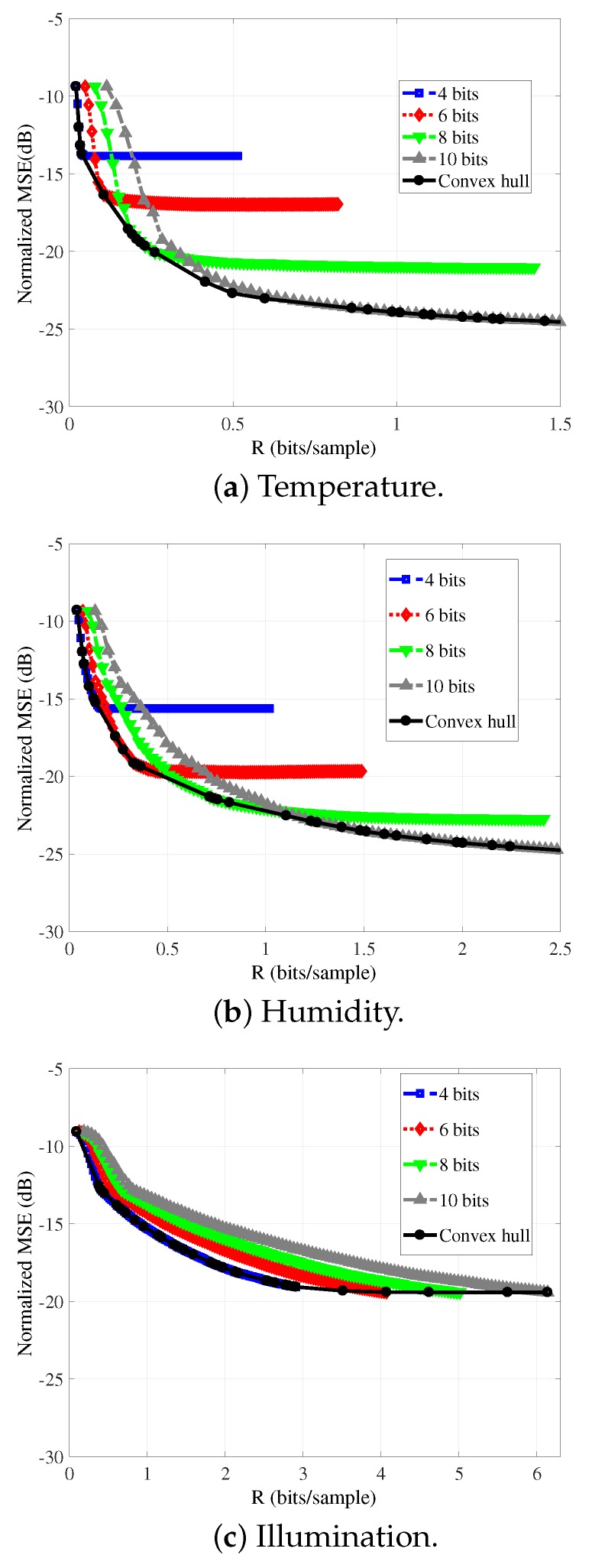
Rate-distortion curves and convex hulls for the reconstruction of signals with LASSO for several quantization bit-depths. The block length *N* is equal to 512.

**Figure 4 sensors-19-00266-f004:**
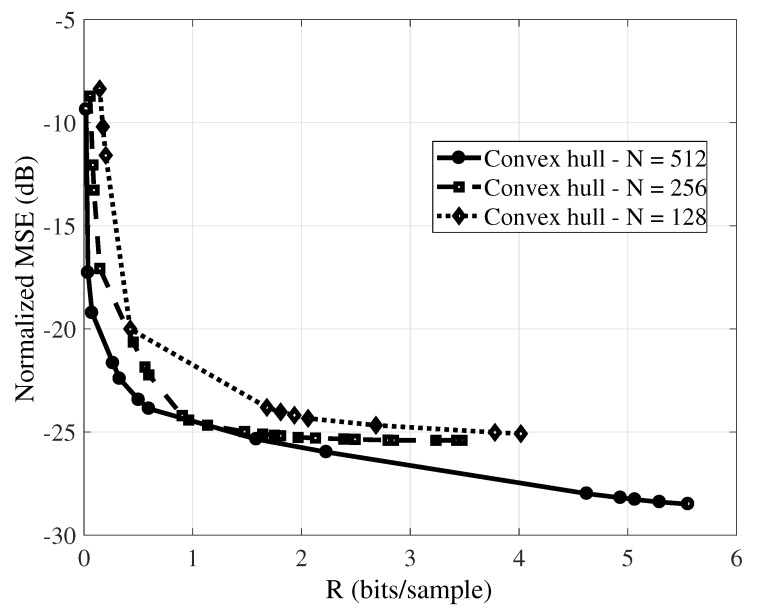
Rate-distortion convex hulls for the reconstruction of temperature signal with LASSO varying the length of monitored signal block (*N*).

**Figure 5 sensors-19-00266-f005:**
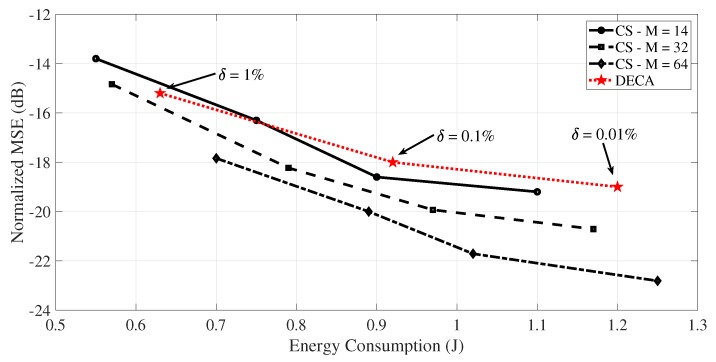
A comparison of the energy consumption between both distributed energy conservation algorithm (DECA) and CS schemes, applied to the reconstruction of N=512 samples of a temperature signal.

**Figure 6 sensors-19-00266-f006:**
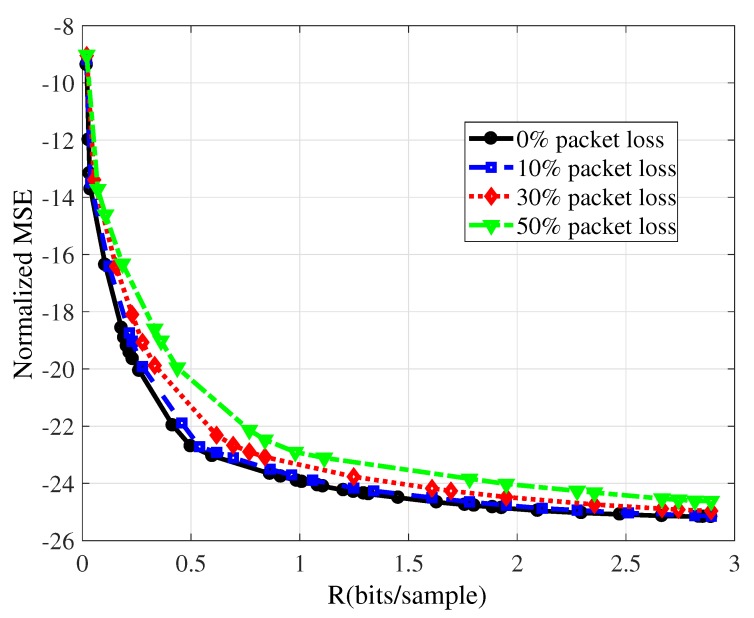
Rate-distortion convex hulls for each packet loss percentage for the reconstruction of N=512 samples of a temperature signal with LASSO, considering the transmission of 1 CS measurement per packet.

**Figure 7 sensors-19-00266-f007:**
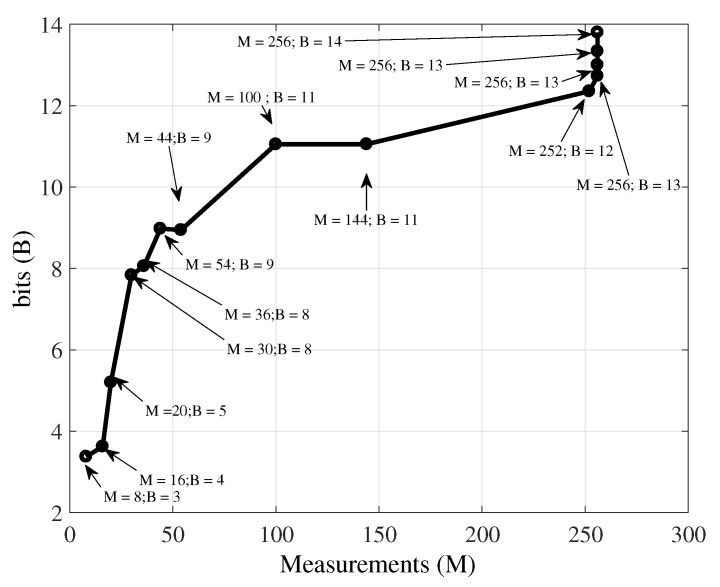
Optimal coding path on the M×B plane, the points that correspond to the RD convex hull, for the reconstruction of a temperature signal with LASSO.

**Figure 8 sensors-19-00266-f008:**
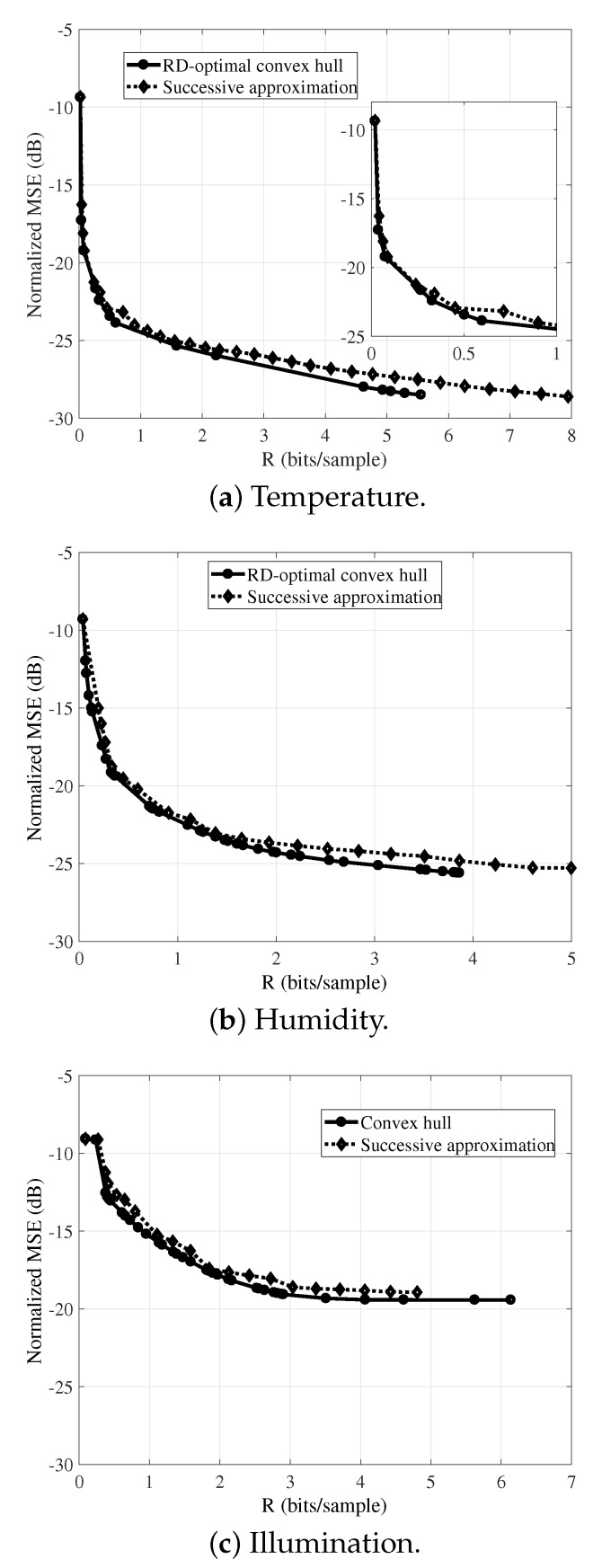
RD-optimal convex hull and rate-distortion curves for the successive approximation scheme, for the reconstruction of the signals with LASSO. The block length *N* is equal to 512.

**Table 1 sensors-19-00266-t001:** Energy consumption of compressive sensing (CS) scheme, applied to the reconstruction of N=512 samples of a temperature signal, using 14 CS measurements.

*B* (bit-depth)	R (bits/sample)	NMSE (dB)	Energy Cons. (J)
4	0.034	−13.9	0.56
6	0.086	−16.3	0.74
8	0.140	−18.6	0.90
10	0.199	−19.2	1.1

**Table 2 sensors-19-00266-t002:** ΔBD between rate-distortion (RD) curves for each packet loss percentage for the reconstruction of N=512 samples of a temperature signal with LASSO, considering the transmission of 1 CS measurement per packet.

Percentage	ΔBD (dB)
0%–10%	0.36
0%–20%	0.64
0%–30%	1.07
0%–40%	1.30
0%–50%	1.73

**Table 3 sensors-19-00266-t003:** Evaluation of the proposed successive approximation scheme capability to save sensor nodes energy—Scenario 1.

$\left(M_{j},B_{j}\right)$	E. Cons. (J)	E. Cons. (J)—App. suc.	Energy Reduction (%)
(8,4)	0.53	0.53	-
(12,6)	1.08	1.03	4.60
(16,8)	1.64	1.44	12.19
(32,10)	2.44	1.88	22.95
(60,11)	3.41	2.37	30.49

**Table 4 sensors-19-00266-t004:** Evaluation of the proposed successive approximation scheme capability to save sensor nodes energy—Scenario 2.

$\left(M_{j},B_{j}\right)$	NMSE (dB)	E. Cons. (J)	E. Cons. (J)—App. suc.	Energy Reduction (%)
(10,4)	−10.27	0.54	0.54	-
(16,6)	−13.97	1.12	1.07	4.46
(32,8)	−16.11	1.73	1.56	9.83
(64,10)	−17.72	2.48	2.07	16.53

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
