# Peer review of "Rate-Distortion Performance and Incremental Transmission Scheme of Compressive Sensed Measurements in Wireless Sensor Networks"

_sensors, 2019, doi:10.3390/s19020266_

Round 1

Reviewer 1 Report

This paper explores the compressive sensing strategy to reduce the number of coefficients to transmit and consequently save the energy of sensor nodes. The paper is easy to follow, but I have several concerns before publishing in the Sensors. 1. The novelty or scientific contribution of this paper is unclear. CS has been widely used in WSN? Therefore, the difference between your work and others should be clarified in the introduction. 2. The literature review problem. More relevant papers about compressive sensing should be cited in the paper, e.g., UL-Isomap based nonlinear dimensionality reduction for hyperspectral imagery classification[J]. ISPRS Journal of Photogrammetry and Remote Sensing, 2014, 89: 25-36. Nonlinear dimensionality reduction via the ENH-LTSA method for hyperspectral image classification[J]. IEEE Journal of Selected Topics in Applied Earth Observations and Remote Sensing, 2014, 7(2): 375-388. 3. The experiment part. The proposed method should be compared with two other similar works in the experiments to fully testify its advantages.

Author Response

[Comment 1:] The novelty or scientific contribution of this paper is unclear. CS has been widely used in WSN? Therefore, the difference between your work and others should be clarified in the introduction.

[Answer/Action:] We included text in the Introduction in order to clarify the contributions of this paper. Please, see the fifth paragraph of Section 1 in the new manuscript. Moreover, Section 2 presents several energy conservation schemes applied to WSN, including CS-based methods. We situate our work within the field in Section 2 and discuss the objectives and contributions of the paper in Subsection 2.1.

[Comment 2:]  The literature review problem. More relevant papers about compressive sensing should be cited in the paper, e.g., UL-Isomap based nonlinear dimensionality reduction for hyperspectral imagery classification[J]. ISPRS Journal of Photogrammetry and Remote Sensing, 2014, 89: 25-36. Nonlinear dimensionality reduction via the ENH-LTSA method for hyperspectral image classification[J]. IEEE Journal of Selected Topics in Applied Earth Observations and Remote Sensing, 2014, 7(2): 375-388.

[Answer/Action:] We included the aforementioned references in the paragraphs 6 and 7 of the Section 2 (new manuscript).

[Comment 3:] The experiment part. The proposed method should be compared with two other similar works in the experiments to fully testify its advantages.

[Answer/Action:] We compare the CS scheme against reference [46] showing the advantages of the CS with respect of the NMSE and energy consumption, and included new results (see Figure 5, lines 7--12 of the second paragraph of Subsection 4.3 and paragraphs 4 and 5 of Subsection 4.3 of the new manuscript). Moreover, we included new results in Subsection 5.3.2 with respect to energy consumption. We included a new experiment showing the advantages of the proposed successive approximation scheme for energy conservation of sensor nodes. Please, see Table 4 and third paragraph of the Subsection 5.3.2 (new manuscript).

Reviewer 2 Report

In the proposed article, the authors consider methods for reducing the amount of transmissions in WSN.
The quantized CS framework is investigated. Incremental CS quantized measurements is discussed. The scheme
allows sensor nodes to first encode signals with a coarse quality and let the sink to ask for refinements.
Compressive sensing framework, with different reconstruction methods are considered. Rate distortion analysis
was done on different quantity and quantizer bit-depths. The robustness against packet loss are proven.

The article is well constructed. Introduction gives a fair background and in chapter 2 the techniques for energy saving in
WSN  are reviewed.
Methods, performance analysis are well done

Author Response

[Comment 1:] The article is well constructed. Introduction gives a fair background and in chapter 2 the techniques for energy saving in WSN  are reviewed.
Methods, performance analysis are well done.

[Answer/Action:] Thank you for reading and evaluating our work and proposal.

Reviewer 3 Report

This paper is about the quantization of compressive sensed measurements in WSNs. The authors described the method and related (previous work) as well as the resulting rate-distortion performance of their proposed distinct reconstruction methods (three different methods) was analyzed for temperature, humidity and illumination data. The paper is of excellent technical depth and presents the problem, proposed solution and representative results to support the scope of this study. Some more results on the energy consuption of the sensors with this scheme would be also advisable to enforce the original algorithm (i.e. energy consumption (vs time alive of sensors) using these CS reconstruction methods and when not)

Author Response

[Comment 1:] Some more results on the energy consumption of the sensors with this scheme would be also advisable to enforce the original algorithm (i.e. energy consumption (vs time alive of sensors) using these CS reconstruction methods and when not).

[Answer/Action:] We included a new experiment showing the advantages of the proposed successive approximation scheme for energy conservation of sensor nodes in a multihop WSN considering the CS-coded framework. Please, see Table 4 and third paragraph of the Subsection 5.3.2 (new manuscript).

Round 2

Reviewer 1 Report

The authors have carefully responded to all my comments. The paper quality has been greatly improved. I think the paper can be published after handling the following minor problems.

the disadvantage or advantage of proposed algorithms should be mentioned in the experimental part.

The Symbols in all the questions should be defined in the paper. The authors are suggested to make a careful checking.

Author Response

[Comment 1:] The disadvantage or advantage of proposed algorithms should be mentioned in the experimental part.

[Answer/Action:] We included text in the lines 5-7 of the last paragraph of Subsection 4.3, and we included a last paragraph in the Subsection 5.3.2, in order to mention the disadvantage / advantage of the proposed CS-based schemes.

[Comment 2:] The Symbols in all the questions should be defined in the paper. The authors are suggested to make a careful checking.

[Answer/Action:] We verified the symbols of the equations, and we included the definition for the Epsilon parameter of eq. (4), in the first line after the eq. (4) - Subsection 3.1.1, and the definition for the Tau parameter of eq. (9), in the first line after the eq. (9) - Subsection 3.1.3.
